# SE(3)-Invariant Multiparameter Persistent Homology for Chiral-Sensitive Molecular Property Prediction

**Andac Demir**
Novartis

**Francis Prael III**
Novartis

**Bulent Kiziltan**
Novartis
{andac.demir, francis.prael_iii, bulent.kiziltan}@novartis.com

## Abstract

In this study, we present a novel computational method for generating molecular fingerprints using multiparameter persistent homology (MPPH). This technique holds considerable significance for key areas such as drug discovery and materials science, where precise molecular property prediction is vital. By integrating SE(3)-invariance with Vietoris-Rips persistent homology, we effectively capture the three-dimensional representations of molecular chirality. Chirality, an intrinsic feature of stereochemistry, is dictated by the spatial orientation of atoms within a molecule, defining its unique 3D configuration. This non-superimposable mirror image property directly influences the molecular interactions, thereby serving as an essential factor in molecular property prediction. We explore the underlying topologies and patterns in molecular structures by applying Vietoris-Rips persistent homology across varying scales and parameters such as atomic weight, partial charge, bond type, and chirality. Our method's efficacy can be further improved by incorporating additional parameters such as aromaticity, orbital hybridization, bond polarity, conjugated systems, as well as bond and torsion angles. Additionally, we leverage Stochastic Gradient Langevin Boosting (SGLB) in a Bayesian ensemble of Gradient Boosting Decision Trees (GBDT) to obtain aleatoric and epistemic uncertainty estimates for gradient boosting models. Using these uncertainty estimates, we prioritize high-uncertainty samples for active learning and model fine-tuning, benefiting scenarios where data labeling is costly or time consuming. Our approach offers unique insights into molecular structure, distinguishing it from traditional single-parameter or single-scale analyses. When compared to conventional graph neural networks (GNNs) which usually suffer from oversmoothing and oversquashing, MPPH provides a more comprehensive and interpretable characterization of molecular data topology. We substantiate our approach with theoretical stability guarantees and demonstrate its superior performance over existing state-of-the-art methods in predicting molecular properties through extensive evaluations on the MoleculeNet benchmark datasets.

## 1 Introduction

Molecular property prediction has received substantial interest in recent years to accelerate the drug discovery process [57] and predict the 3D structure of proteins [34], with models showing potential to solve contemporary problems in materials science [51] and quantum chemistry [15]. Recent adaptations of GNNs, including GIN [65, 49], GAT [61, 63], and MPNN [67], have demonstrated leading performance in molecular representation learning, excelling across various compound datasets

NeurIPS 2023 AI for Science Workshop.

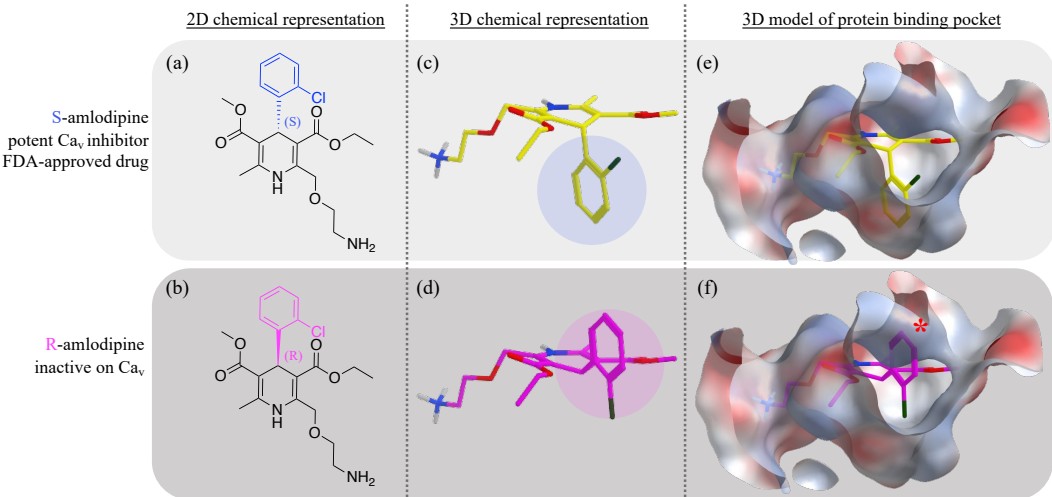

Figure 1: **Chirality** (a, b) 2D chemical representation of (a) S-amlodipine, an FDA-approved calcium channel inhibitor and its enantiomer (b) R-amlodipine, a compound lacking calcium channel activity. The stereocenters on each are labeled in blue and magenta, respectively. (c, d) 3D representations of these molecules, with differences in 3D space highlighted. (e) Model of S-amlodipine bound within its binding pocket on a calcium channel, generated from empirical structural data (PDB ID: 7JPX [24]). (f) A model of R-amlodipine bound to the same pocket. The red asterisk marks a steric clash between the molecule and the binding pocket, indicating the inability of R-amlodipine to bind this pocket in this orientation. The inability of R-amlodipine to form a productive binding interaction is consistent with its lack of biological activity on this channel.

for bioactivity and molecular property prediction tasks. Despite their success, utilization of GNNs for molecular property prediction suffers from 3 severe limitations:

*Oversmoothing:* Neighborhood information is typically aggregated by permutation invariant, but non-injective operations such as an average, sum or max. This leads to an *oversmoothing* problem, where node embeddings converge to similar values and the information-to-noise ratio of the message received by the nodes decreases [9, 47].

*Oversquashing:* GNNs are susceptible to a bottleneck known as *oversquashing*. This occurs when the amount of information aggregated from an exponentially growing receptive field exceeds a node's capacity to process. As the number of GNN layers (and thus the receptive field) increases, all the information is compressed into fixed-length node vectors, potentially causing the loss of important information from more distant nodes [4].

*Capturing 3D spatial information or conformational changes:* Traditional GNNs primarily focus on atomic connectivity through ionic and covalent bonds, often neglecting crucial intermolecular forces like hydrogen bonds, dipole-dipole interactions, and Van der Waals forces. These forces, although subtle, significantly impact molecular conformation and the resulting physical, chemical, and reactive properties of molecules. Despite their seemingly weak nature, the incorporation of 3D molecular conformers into analysis improves the accuracy of molecular property prediction [52, 27, 38, 25]. Yet, generating low energy stable 3D conformers for large-scale applications is computationally demanding [66, 53, 23]. Some methods instead use bond lengths, angles, or torsion angles as additional 3D features [10, 27, 25].

Chirality is a concept in stereochemistry, a branch of chemistry focused on the 3D arrangement of atoms in molecules, and the effects of these arrangements on the chemical properties and reactions of those molecules. A molecule is said to be chiral if it cannot be superimposed on its mirror image [8]. It is paramount in drug design as it can lead to chiral sets of molecules, despite their similar physicochemical properties, exhibiting significantly different affinities, efficacies, and potencies when interacting with drug targets [55]. An example of this is the common cardiovascular medication, amlodipine (Figure 1) [19]. Amlodipine, functioning by inhibiting voltage-gated calcium channels (VGCCs)[5], exists as two chiral forms: $S$-amlodipine and $R$-amlodipine, with the former being therapeutically active and the latter mostly inactive [30]. Despite their similar chemical structures, the distinct 3D orientations allow $S$-amlodipine to bind to a VGCC pocket and inhibit its activity, while preventing $R$-amlodipine from doing the same [24]. Consequently, $S$-amlodipine exhibits significant

therapeutic value, unlike $R$-amlodipine. This underscores the necessity to consider chirality in drug design for achieving high specificity and minimal side effects.

One of the most common manifestations of chirality in organic molecules is tetrahedral (point) chirality, where a central atom, typically carbon, is bonded to four non-equivalent chemical groups. This arrangement results in non-superimposable mirror images called enantiomers [8]. Enantiomers are denoted by dashed (Figure 1a) and bold (Figure 1b) wedges, indicating the orientation of the bonds relative to the plane of the molecule. Accurately differentiating enantiomers poses a challenge for Euclidean group [E(3)]-invariant GNNs that solely consider pairwise atomic distances or bond angles in their message updates, like SchNet [52]. These models struggle to distinguish between enantiomers due to the inversion of chiral centers upon reflection. To address this, incorporating Special Euclidean group [SE(3)]-invariance becomes crucial in molecular property prediction [2, 39]. By accounting for SE(3)-invariance, models can effectively generalize across various molecular conformations, including chiral systems, leading to enhanced performance even with limited training data.

**The key contributions of this paper are:**

1. We introduce a novel compound fingerprinting method by integrating SE(3)-invariance into *Vietoris-Rips persistent homology*, generating robust and versatile representations for chiral compound property prediction, thereby eliminating the need for multi-conformer data augmentation and expensive equivariant operations, such as spherical harmonics and Clebsch-Gordan coefficients.

2. We leverage Stochastic Gradient Langevin Boosting (SGLB) [60] in a Bayesian ensemble of Gradient Boosting Decision Trees (GBDT) [41]. This allows us to separately quantify both *aleatoric* and *epistemic uncertainties*, aiding in error prevention and the identification of informative compounds for data collection.

3. We establish *theoretical guarantees for the stability of compound fingerprints* derived through MPPH.

4. Our method, validated through empirical evaluations on MoleculeNet benchmark datasets, surpasses state-of-the-art baselines significantly, proving the *superior predictive performance* of the MPPH-based approach.

## 2 Related Work

There are two primary strategies for predicting the physical and chemical properties of molecules: 1) The utilization of established models such as random forest or gradient boosting decision trees, which rely on expert-engineered descriptors or molecular fingerprints, and 2) The optimization of GNN model architectures. In the first strategy, models operate on molecular fingerprints like SMILES [71], Dragon descriptors [43], Extended Connectivity Fingerprints (ECFP) [50, 70] or eigenspectrum of Coulomb matrices [46]. Enhancements in this approach can be achieved by enriching the feature representation of nodes (atoms) with additional chemical information. Additionally, some studies have used explicit 3D atomic coordinates to further improve performance [52, 36, 18, 20]. The second strategy emphasizes optimizing model architecture and enhancing neighborhood aggregation. Graph Convolutional Neural Networks (GCN), for instance, generate a compound's feature representation through the convolution operations performed in the spectral domain of the compound's $2D$ graph [64], which is obtained by transforming the graph into a set of eigenvectors and eigenvalues. Similarly, Message Passing Neural Networks (MPNN)[29] update node and edge representations by passing messages between nodes, iteratively constructing a graph representation. Recently, conformer generation, accounting for molecules' 3D structures, has become crucial in property prediction [7]. This is particularly important for non-rigid molecules that can adopt diverse conformations under varying conditions. 3D Infomax [56] pre-trains a $2D$ network by maximizing the mutual information (MI) between its representation of a molecular graph and a 3D representation produced from the molecules' conformers. The weights of $2D$ network are then fine-tuned to predict properties.

Additionally, the development of chiral-sensitive molecular representations is an area of significant interest in computational chemistry. DimeNet [27] and its faster successors DimeNet++ [26] and GemNet [25] use spherical harmonics and Clebsch-Gordan coefficients to represent the relative directional information. SphereNet [38] improves computational efficiency by proposing spherical

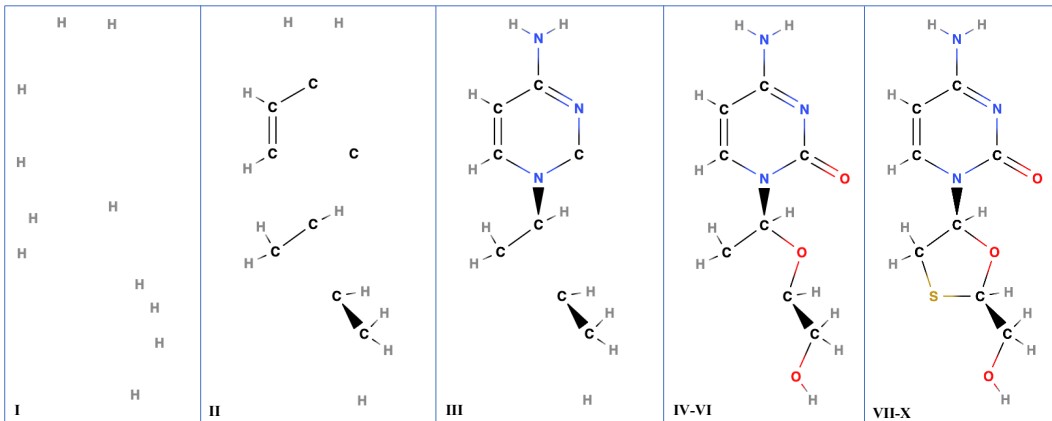

Figure 2: **Graph Decomposition** involves masking certain vertices based on their parameter values, and then considering only the remaining vertices and edges. In the given dataset, 10 unique atoms (H, C, N, O, F, P, S, Cl, Br, I) are identified, leading to the creation of 10 subgraphs. The atoms are color-coded as follows: Gray for Hydrogen, Black for Carbon, Blue for Nitrogen, Red for Oxygen, and Yellow for Sulfur. Note that some subgraphs may be identical. The original compound, lamivudine (a hepatitis B antiviral), is displayed in the last column. Vertices with the highest atomic mass are masked first and added in ascending order of atomic mass. Figure 2 illustrates the graph evolution along the y-axis in the initial column of Figure 6. Our framework builds the Vietoris-Rips complexes for each subgraph, as shown in the rows of Figure 6, and computes the rank of the homology groups of dimensions 0 and 1 for each sequence of simplicial complexes. The process is then repeated using different parameters: partial charge (in ascending order of decile groups in the partial charge histogram) and bond types (starting with vertices forming a ring structure, then adding vertices connected by triple, double, and finally, single bonds).

message passing, while SE(3)-Transformer[22] leverages Wigner-D matrices for learning SO(3)'s irreducible representations. While these models can potentially learn chirality, their efficacy for molecular property prediction remains unexplored. Moreover, they utilize computationally intensive equivariant operations, such as spherical harmonics and Clebsh-Gordan coefficients [59, 22]. This introduces increased computational complexity and high-dimensional representations, thereby complicating model implementation and optimization.

In this paper, we build upon our previous work, ToDD [13, 14], enhancing its utility in ligand-based virtual screening and molecular property prediction. We generate topological fingerprints that remain invariant under SE(3) group actions, accurately representing relative atomic arrangements and tetrahedral chiral configurations.

## 3   Invariance of Persistent Homology Under $E(3)$ Transformations

We present the concept of multiparameter persistent homology as a three-step process in Section C. Firstly, the process involves 'graph decomposition' (Figure 2), where vertices are masked based on their values in either ascending or descending order, subsequently breaking down a larger graph into smaller subgraphs. Secondly, 'persistent homology' comes into play, tracking changes in topological features like birth and death times within each subgraph's sequence of simplicial complexes. Lastly, in the 'vectorization' step, these records are converted into a vector, a form that can be readily used in machine learning models.

In this section, we establish the invariance of persistent homology induced by Vietoris-Rips filtration under $E(3)$. We first present a lemma related to the preservation of pairwise distances under $E(3)$ transformations and then prove a theorem that demonstrates the $E(3)$ invariance of Vietoris-Rips persistent homology.

**Lemma 3.1.** *Preservation of Pairwise Distances under* $E(3)$*: Let $X \subset \mathbb{R}^3$ be a finite set of points, and let $g \in E(3)$ be an Euclidean transformation. Then, for any two points $x_i, x_j \in X$, the pairwise distance between $x_i$ and $x_j$ is preserved under the action of g, i.e., $\|g(x_i) - g(x_j)\| = \|x_i - x_j\|$.*

*Proof.* Let $g$ be an Euclidean transformation represented by a rotation matrix $R \in SO(3)$, a translation vector $t \in \mathbb{R}^3$, and a reflection matrix $M \in O(3) \setminus SO(3)$. Then, for any two points $x_i, x_j \in X$,

$$\|g(x_i)-g(x_j)\| = \|M(Rx_i+t)-M(Rx_j+t)\| = \|M(R(x_i-x_j))\| = \|R(x_i-x_j)\| = \|x_i-x_j\|. \tag{1}$$

The second equality holds because reflections preserve distances, and the third equality holds because rotations preserve distances. □

**Theorem 3.2.** $E(3)$ *Invariance of Vietoris-Rips Persistent Homology: Let $X \subset \mathbb{R}^3$ be a finite set of points, and let $g \in E(3)$ be an Euclidean transformation. Then, the Vietoris-Rips persistent homology of $X$ is invariant under the action of $g$, i.e., $PH(VR_\epsilon(X)) \cong PH(VR_\epsilon(g(X)))$ for all $\epsilon > 0$.*

*Proof.* Let $X' = g(X)$. By Lemma 3.1, the pairwise distances between points in $X$ are preserved under the action of $g$. Therefore, for any $\epsilon > 0$, a simplex $\sigma$ is included in $VR_\epsilon(X)$ if and only if the corresponding simplex $g(\sigma)$ is included in $VR_\epsilon(X')$. This implies that the simplicial complexes $VR_\epsilon(X)$ and $VR_\epsilon(X')$ are isomorphic for all $\epsilon > 0$, and consequently, their homology groups are isomorphic as well. Since the persistent homology is derived from the homology groups of the Vietoris-Rips complexes for all $\epsilon > 0$, it follows that $PH(VR_\epsilon(X)) \cong PH(VR_\epsilon(g(X)))$. □

## 4 Incorporating SE(3)-Invariance into Multiparameter Persistent Homology

In the chiral-sensitive property prediction task, the filtration function $f$ assigns to each atom a value that represents whether it's a chiral center and, if so, what type of chiral center it is. Let $M$ denote a molecule, modeled as a connected graph where vertices represent atoms and edges represent bonds. Denote the vertex set of $M$ as $V(M)$. For each vertex $v \in V(M)$, let $C(v)$ be the set of its neighboring vertices, ordered according to the Cahn-Ingold-Prelog (CIP) priority rules. We define a chiral center as a vertex $v$ with four different groups attached, i.e., $|C(v)| = 4$ and all elements of $C(v)$ are distinct. The configuration of a chiral center $v$, determined by the CIP rules, is denoted as:

$$\text{config}(v) = \begin{cases} 1 & \text{if the order of elements in } C(v) \text{ is clockwise,} \\ 2 & \text{if the order of elements in } C(v) \text{ is counterclockwise.} \end{cases}$$

The filtration function $f : V(M) \to \{0, 1, 2\}$ can now be defined compactly as:

$$f(v) = \begin{cases} 0 & \text{if } |C(v)| \neq 4, \\ \text{config}(v) & \text{otherwise.} \end{cases}$$

For each value $r \in \mathbb{R}$, the sublevel set $X_r = f((-\infty, r])$ is the set of all atoms with $f$-values less than or equal to $r$. The nested sequence of sublevel sets $X_{r_1} \subseteq X_{r_2} \subseteq \cdots$ induces a nested sequence of Vietoris-Rips complexes $VR_\epsilon(X_{r_1}) \subseteq VR_\epsilon(X_{r_2}) \subseteq \cdots$. If we build a VR complex and compute the persistent homology based on this filtration function, we obtain a characterization of the molecule that identifies and distinguishes its chiral centers. Importantly, this characterization depends only on the relative configuration of the atoms, not on the absolute position/orientation of the molecule. Therefore, it's expected to be invariant under SE(3) transformations.

The persistent homology of this nested sequence of complexes is a sequence of homology groups $PH(VR_\epsilon(X_{r_1})), PH(VR_\epsilon(X_{r_2})), \ldots$ and homomorphisms between them, which capture the topological features (connected components, loops, and voids) of the molecule that persist across multiple scales $r_1, r_2, \ldots$.

**Theorem 4.1.** *The Vietoris-Rips complex and its persistent homology are invariant under SE(3) transformations, assuming that the filtration function, $f$, is also SE(3)-invariant.*

*Proof.* Let $T : \mathbb{R}^3 \to \mathbb{R}^3$ be a transformation in SE(3), let $X \subseteq \mathbb{R}^3$ be the set of atoms in the molecule, and let $X' = T(X)$ be the transformed set of atoms. Let $f$ be the filtration function, and let $f' = f \circ T$ be the transformed filtration function. Note that $f'$ assigns to each atom in $X'$ the same value that $f$ assigns to the corresponding atom in $X$, because $f$ is assumed to be SE(3)-invariant.

First, we show that the Vietoris-Rips complex is invariant under $T$. For any $\epsilon > 0$ and $r \in \mathbb{R}$, we have $VR_\epsilon(X_r) = VR_\epsilon(X'_r)$, where $X_r$ and $X'_r$ are the sublevel sets of $f$ and $f'$ at $r$, respectively. This is because for any two atoms $x, y \in X_r$ with $d(x, y) < \epsilon$, their transformed counterparts $x' = T(x), y' = T(y) \in X'_r$ also satisfy $d(x', y') < \epsilon$. This is a result of the fact that $T$ is an

isometry (i.e., it preserves distances), which is a property of all transformations in SE(3). Therefore, any simplex in $VR_\epsilon(X_r)$ corresponds to a simplex in $VR_\epsilon(X'_r)$, and vice versa.

Next, we show that the persistent homology is also invariant under $T$. The persistent homology is computed from the nested sequence of Vietoris-Rips complexes, which we have shown to be invariant under $T$. Furthermore, the homomorphisms between the homology groups in the sequence are induced by the inclusions $X_{r_1} \subseteq X_{r_2} \subseteq \cdots$, which correspond under $T$ to the inclusions $X'_{r_1} \subseteq X'_{r_2} \subseteq \cdots$. Therefore, the entire structure of the persistent homology, including both the homology groups and the homomorphisms between them, is preserved under $T$. $\qquad\square$

**Theorem 4.2.** *Assuming that the filtration function, $f$, accurately identifies the chiral centers and their configurations in a compound, and is non-invariant under reflection, the Vietoris-Rips Persistent Homology, which operates on the nested sequence of subgraphs induced by $f$, is also non-invariant under reflection.*

*Proof.* Consider a molecule $M$ and its reflection $\rho(M)$. By the definition of $f$, $f(v) \neq f(\rho(v))$ for any chiral center $v$ with 'R' or 'S' configuration due to the inversion of the configuration under reflection. Denote by $VR_f(M)$ and $VR_f(\rho(M))$ the filtered Vietoris-Rips complexes of $M$ and $\rho(M)$ with the filtration function $f$. Then, due to the difference in $f$ values, $VR_f(M) \not\cong VR_f(\rho(M))$. As persistent homology is derived from the filtered Vietoris-Rips complex, the non-isomorphism of the complexes implies non-isomorphism of the corresponding homologies. $\qquad\square$

## 4.1 Stability of MPPH Fingerprints

Stability in single parameter persistence vectorizations pertains to the consistency of the mapping from the space of persistence diagrams to the space of functions or vectors, assessed using the Wasserstein distance metric [1, 54]. Essentially, minor changes in the persistence diagram shouldn't drastically alter the vectorization. This concept has been leveraged to demonstrate the stability of Multiparameter Persistent Homology (MPPH) Fingerprints, where changes in the vectorizations are bounded by changes in the corresponding persistence diagrams. The induced matching distance between multiple persistence diagrams (Equation 3) and the distance between induced MPPH Fingerprints (Equation 4) are defined to uphold this stability. For more in-depth explanation and proof of the following theorem, please refer to D.

**Theorem 4.3.** *If $\varphi$ is a stable, single-parameter persistence vectorization, the resulting MPPH Fingerprint $\mathcal{M}\varphi$ is also stable. That is, for any pair of graphs $\mathcal{G}^+$ and $\mathcal{G}^-$, a certain constant $\widehat{C}\varphi > 0$ exists, ensuring the following inequality:*

$$\mathfrak{D}(\mathbf{M}_\varphi(\mathcal{G}^+), \mathbf{M}_\varphi(\mathcal{G}^-)) \leq \widehat{C}_\varphi \cdot \mathbf{D}_{p_\varphi}(\{PD(\mathcal{G}^+)\}, \{PD(\mathcal{G}^-)\})$$

# 5 Experiments

We thoroughly evaluate the performance of our methods against the 8 state-of-the-art baselines: GraphConv [16], Weave [35], SchNet [52], Node-MPN [29], D-MPNN [67, 68], MGCN [40], GRAPHCL [69] and 3D Infomax [56] on 6 benchmark datasets (Lipophilicity, FreeSolv, ESOL, BACE, BBBP, ClinTox), primarily sourced from MoleculeNet [64] (a large scale benchmark for molecular machine learning) and adopt the 80/10/10 scaffold splits provided by OGB [32]. See Section G.1 for further details.

## 5.1 Uncertainty Quantification via SGLB Ensembles

The gradient boosting algorithm [21], enhanced by Stochastic Gradient Langevin Boosting (SGLB), iteratively builds a model $F$ to minimize empirical risk $L(F|D)$, where $D$ is the set of $N$ data points $(x_1, y_1), (x_2, y_2), ..., (x_N, y_N)$. In this context, $x$ represents the MPPH Fingerprint given in tabular data form, and $y$ is the associated property. The model update equation is $F^{(t)}(x) = F^{(t-1)}(x) + \eta h^{(t)}(x, \phi^{(t)})$, where $F^{(t-1)}$ is the prior model, $h^{(t)}$ is a selected weak learner, $\phi^{(t)}$ is the model parameters, and $\eta$ is the learning rate. SGLB creates an ensemble of models, estimating uncertainty through the entropy of their predictive distributions. Bayesian inference approximates the true posterior $p(\theta|D)$ of this ensemble, with total uncertainty, *aleatoric uncertainty* (knowledge uncertainty), and expected *episdemic uncertainty* (data uncertainty) estimated via entropy and variance

calculations. In the case of continuous-valued targets, estimates are obtained using the law of total variance. To reach a globally optimal solution, SGLB injects Gaussian noise into the gradients and introduces a regularization term that moderates the influence of preceding models. For detailed mathematical explanations, refer to F.1, and for implementation specifics, see F.2.

## 5.2 Experimental Results

We adopt a GBDT model, leveraging SGLB optimization with SE(3)-invariant multiparameter persistent homology (MPPH) Fingerprints as the input feature set. The model, comprising 1000 boosting stages, is trained with a refined maximum tree depth of 6, ascertained through exhaustive hyperparameter tuning. A learning rate of 0.05 is employed, and the criterion for splitting a node within the tree is stipulated as a minimum of 2 samples. Our method surpasses contemporary state-of-the-art GNNs as well as conventional Random Forest (RF) models, trained on ECFP-4 fingerprints as shown in Table 1.

Our SE(3)-invariant MPPH methodology outperforms top MoleculeNet models on FreeSolv [45], ESOL [12], BACE [58], BBBP [42], ClinTox [28], while demonstrating comparable results on Lipophilicity. These outcomes suggest that MPPH not only exceeds the performance of the leading MoleculeNet models but does so without necessitating the training of large-scale GNNs or generating 3D conformations. Moreover, our method displays a *marked improvement* over all GNN baselines on FreeSolv, ESOL, and ClinTox. We postulate two reasons for these superior results. Firstly, enantiomers, despite sharing similar physiochemical properties, can display drastically different biological activities. Hence, our approach is expected to excel in biological systems such as drug potency (BACE), blood-brain barrier penetration (BBBP), and clinical trial toxicity (ClinTox) compared to those evaluating physiochemical properties like lipid solubility (Lipophilicity), and aqueous solubility (FreeSolv and ESOL). BBBP might represent a balanced case between physiochemical and biological factors [48]. Secondly, the performance of GNNs can be affected worse on smaller datasets like FreeSolv and ESOL due to the restricted number of training samples. The limited size of datasets (up to $\sim 1000$ training molecules) negatively affects the performance of GNNs due to data sparsity [67]. Furthermore, MPPH can effectively leverage domain knowledge to improve property prediction scores. For instance, it is a commonly acknowledged principle that the introduction of nonpolar groups, such as methyl groups, into a molecule can elevate its lipophilicity. MPPH, by incorporating bond polarity as an auxiliary parameter, effectively integrates crucial domain information, thereby augmenting the performance of lipophilicity prediction.

Table 1: SE(3)-Invariant MPPH vs. state-of-the-art baselines. Shown is the root mean squared error (RMSE) for Lipophilicity, FreeSolv, ESOL (lower is better), and the area under the ROC-curve (ROC-AUC) for BACE, BBBP, ClinTox (higher is better). Best in **bold**, top baseline underlined. Improvement and deterioration against top baseline are color-coded.

| Model | Lipophilicity $\downarrow$ | FreeSolv $\downarrow$ | ESOL $\downarrow$ | BACE $\uparrow$ | BBBP $\uparrow$ | ClinTox $\uparrow$ |
|---|---|---|---|---|---|---|
| ECFP-4+RF | 0.706±0.011 | 0.560±0.066 | 1.399±0.177 | 0.881±0.027 | 0.924±0.024 | 0.859±0.023 |
| GraphConv | 0.712±0.049 | 2.900±0.135 | 1.068±0.050 | 0.854±0.011 | 0.877±0.036 | 0.845±0.051 |
| Weave | 0.813±0.042 | 2.398±0.250 | 1.158±0.055 | 0.791±0.008 | 0.837±0.065 | 0.823±0.023 |
| SchNet | 0.909±0.098 | 3.215±0.755 | 1.045±0.064 | 0.750±0.033 | 0.847±0.024 | 0.717±0.042 |
| D-MPNN | **0.646±0.041** | 1.010 ± 0.064 | 0.980±0.258 | 0.878±0.032 | 0.913±0.026 | 0.894±0.027 |
| MGCN | 1.113±0.041 | 3.349±0.097 | 1.266±0.147 | 0.734±0.030 | 0.850±0.064 | 0.634±0.042 |
| Node-MPN | 0.672±0.051 | 2.185±0.952 | 1.167±0.430 | 0.815±0.044 | 0.913±0.041 | 0.879±0.054 |
| GRAPHCL | 0.714±0.011 | 3.744±0.292 | 0.959±0.047 | 0.772±0.040 | 0.711±0.020 | 0.511±0.055 |
| 3D Infomax | 0.695±0.012 | 2.337±0.227 | 0.894±0.028 | 0.794±0.019 | 0.691±0.011 | 0.594±0.032 |
| **SE(3)-I MPPH** | 0.738±0.025 | **0.354±0.053** | **0.612±0.083** | **0.897±0.012** | **0.940±0.021** | **0.993±0.004** |
| Relative gains | - | -36.8% | -31.5% | +0.016 | +0.016 | +0.099 |

In highly imbalanced datasets such as BBBP and ClinTox, the ROC-AUC score provides an overly optimistic view of the model's performance, as the false positive rate does not change significantly with different thresholds in the presence of a large number of negative samples. To get a more comprehensive assessment of the model's performance on these datasets, we also present the PRC-AUC and F1-scores in Section G.3. In contrast to the RF model trained on ECFP-4 fingerprints, which significantly overfits to the majority class (negative labels), MPPH displays superior performance. Its PRC-AUC and F1-score metrics particularly highlight MPPH's proficiency in handling class imbalance, notably in the ClinTox dataset. Figure 3 illustrates the combined aleatoric and epistemic

uncertainty, as derived from SGLB, associated with regression predictions. Additional details on quantifying uncertainty in property predictions from classification models can be found in G.4.

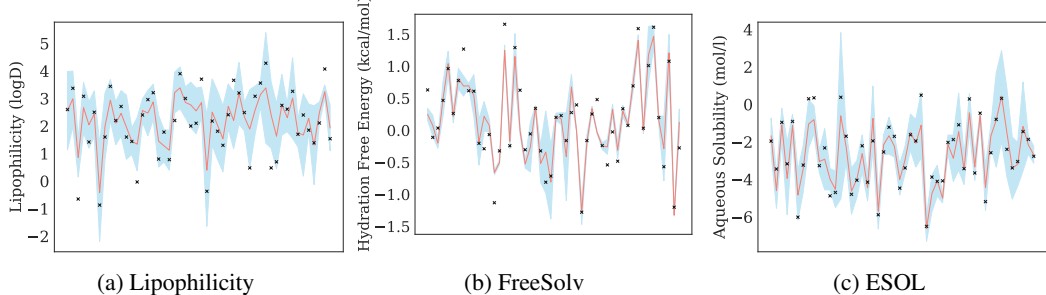

|     (a) Lipophilicity     |     (b) FreeSolv     |     (c) ESOL     |

Figure 3: The red curve signifies the mean prediction, and the surrounding blue band represents the total uncertainty around the mean. The inference of our model appears to be effectively rational, as the congruity between both the mean and its associated uncertainty provides an apt fit to the data.

In our ablation study, as detailed in Table 2, we demonstrate that incorporating domain-specific features, specifically SE(3)-invariance and chirality, significantly improves our model's performance across various metrics. We explore the impact of single-parameter persistence, focusing on factors like atomic mass, partial charge, bond type, and notably chirality, each influencing model effectiveness uniquely. Our approach combines Betti vectorizations from these parameters, including chirality and SE(3)-invariance, creating a holistic representation. This integration of orthogonal and complementary data enhances the model's ability to accurately depict molecular structures, thereby improving predictive accuracy in a chemically meaningful manner. The inclusion of SE(3)-invariance, in particular, aligns the model with the inherent symmetries of three-dimensional space, a key aspect for modeling molecular interactions. This feature ensures consistent model predictions, independent of the molecule's spatial orientation or position.

Table 2: **Comparative Analysis of SE(3)-Invariant MPPH, MPPH, and Single Parameter Persistent Homology.** This figure displays the performance contrasts among SE(3)-Invariant MPPH (depicted in the last column), MPPH (represented in the penultimate column), and Single Parameter Persistent Homology. The superior performance across different metrics and datasets is indicated in **bold** and highlighted. The juxtaposition first provides the efficacy and robustness of multiparameter persistent homology, and then highlights the superior performance of SE(3)-Invariant MPPH.

| Dataset | Metric | Atomic Mass (AM) | Partial Charge (PC) | Bond Type (BT) | Chirality | AM+PC+BT | All Params. |
|---|---|---|---|---|---|---|---|
| **Lipophilicity** | RMSE ↓ | 1.018 | 1.121 | 1.028 | 1.194 | 0.765 | **0.738** |
| **FreeSolv** | RMSE ↓ | 0.542 | 0.427 | 1.107 | 0.773 | 0.378 | **0.354** |
| **ESOL** | RMSE ↓ | 0.991 | 0.977 | 1.339 | 1.496 | 0.624 | **0.612** |
| **BACE** | ROC-AUC ↑ | 0.833 | 0.852 | 0.840 | 0.802 | 0.885 | **0.897** |
| **BACE** | PRC-AUC ↑ | 0.748 | 0.829 | 0.808 | 0.757 | 0.880 | **0.905** |
| **BACE** | F1 ↑ | 0.682 | 0.762 | 0.828 | 0.778 | 0.800 | **0.815** |
| **BBBP** | ROC-AUC ↑ | 0.876 | 0.828 | 0.910 | 0.810 | 0.928 | **0.940** |
| **BBBP** | PRC-AUC ↑ | 0.956 | 0.923 | 0.975 | 0.920 | 0.978 | **0.987** |
| **BBBP** | F1 ↑ | 0.924 | 0.901 | 0.921 | 0.890 | 0.920 | **0.936** |
| **ClinTox** | ROC-AUC ↑ | 0.693 | 0.702 | 0.986 | 0.735 | 0.988 | **0.993** |
| **ClinTox** | PRC-AUC ↑ | 0.318 | 0.182 | 0.896 | 0.294 | 0.900 | **0.923** |
| **ClinTox** | F1 ↑ | 0.296 | 0.091 | 0.857 | 0.231 | 0.864 | **0.870** |

# 6 Conclusion

We have developed a novel method that employs multiparameter persistent homology for molecular fingerprinting. This method adeptly captures tetrahedral chirality by combining SE(3)-invariance with Vietoris-Rips persistent homology. This produces topological descriptors of molecules that depend only on the relative atomic configurations, not on the absolute position/orientation of the molecule. Paired with a Bayesian ensemble, our method estimates both epistemic and aleatoric uncertainties. Our approach is backed by theoretical stability guarantees, and outperforms GNN variants on the MoleculeNet benchmark datasets.

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

# Appendix

## A Stereochemistry and Special Euclidean Geometry in the Context of Molecular Property Prediction

In this paper, we explore the rigorous connection between stereochemistry and Special Euclidean geometry [SE(3)] in the context of molecular property prediction. We provide mathematical definitions, lemmas, theorems, and proofs to elucidate the relationship between the 3D arrangement of atoms in molecules and their properties. Our findings have significant implications for the development of new SE(3)-based molecular representations and algorithms that can more effectively exploit the 3D geometric information of molecules, leading to improved prediction of their properties and interactions. Stereochemistry is concerned with the study of molecules in 3D space. The spatial arrangement of atoms in a molecule is known as its conformation. We give the definitions of some key concepts in stereochemistry and Euclidean geometry in A.1.

### A.1 Preliminaries

**Definition A.1.** *Isomers are molecules with the same molecular formula but different arrangements of atoms in space.*

**Definition A.2.** *Chirality is a property of a molecule that makes it non-superposable on its mirror image.*

**Definition A.3.** *A molecule is a set of atoms $G = a_1, a_2, \ldots, a_n$, where each atom $a_i$ has a unique identifier and a position in 3D space denoted by the vector $\mathbf{x}_i \in \mathbb{R}^3$.*

**Definition A.4.** *A molecular conformation $C$ is a set of positions $\mathbf{x}_1, \mathbf{x}_2, \ldots, \mathbf{x}_n$ representing the arrangement of atoms in a molecule $M$.*

**Definition A.5.** *Orthogonal Group, $O(3)$, is the set of all $3 \times 3$ orthogonal matrices, i.e., the set of all real matrices $A \in \mathbb{R}^{3 \times 3}$ that satisfy the condition $AA^T = A^T A = I$, where $A^T$ denotes the transpose of $A$, and $I$ is the $3 \times 3$ identity matrix.*

*$A$ represents linear transformations that preserve the lengths of vectors and the angles between them. In the context of Euclidean transformations, elements of $O(3)$ are used to represent rotations and reflections in 3D space.*

**Definition A.6.** *Special Orthogonal Group, $SO(3)$, is the set of all $3 \times 3$ orthogonal matrices with determinant equal to 1, i.e., the set of all real matrices $A \in \mathbb{R}^{3 \times 3}$ that satisfy the conditions $AA^T = A^T A = I$ and $\det(A) = 1$.*

*$SO(3)$ is a subgroup of the orthogonal group $O(3)$, and represents the set of all proper rotations in 3D space, i.e., these transformations do not involve reflections.*

**Definition A.7.** *Euclidean transformations, $E(3)$, are transformations that preserve distances in 3D space. They can be represented as a combination of a rotation matrix $R \in SO(3)$, a translation vector $t \in \mathbb{R}^3$, and a reflection matrix $M \in O(3) \setminus SO(3)$.*

**Definition A.8.** *Special Euclidean group, SE(3), is a group of rigid transformations in 3D space, which includes translations and rotations, and is defined as $SE(3) = T(\mathbf{R}, \mathbf{t}) | \mathbf{R} \in SO(3), \mathbf{t} \in \mathbb{R}^3$, where $SO(3)$ is the group of all rotation matrices $\mathbf{R}$ and $\mathbf{t}$ is a translation vector. Invariance under SE(3) transformations means that a property doesn't change under any rotation or translation of the entire system in 3D space, but varies with reflection.*

### A.2 Stereochemistry and SE(3)

In this section, we establish the link between stereochemistry and SE(3) geometry by examining the effect of rigid transformations on molecular conformations. We first present a corollary related to molecular isomorphism and then prove a theorem that connects molecular chirality to the concept of orientation preservation in SE(3) geometry.

**Theorem A.1.** *The relative arrangement of atoms in a molecule $G$ is preserved under any rigid transformation $T(\mathbf{R}, \mathbf{t}) \in SE(3)$.*

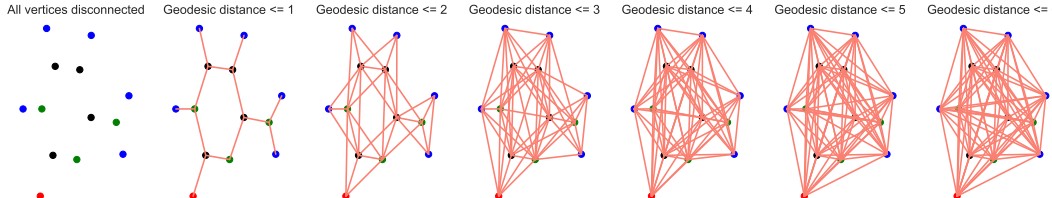

Figure 4: **Vietoris-Rips Filtration and Topological Evolution of Cytosine Molecule.** The subplots illustrate the gradual connection of the molecule's atoms based on the geodesic distance between them. Starting with all vertices (atoms) disconnected, each subsequent plot connects atoms within an increasing geodesic distance. The atoms are color-coded based on their atomic number: Hydrogen (blue), Carbon (black), Nitrogen (green), and Oxygen (red). This visualization reveals the topological changes in the molecule's structure.

*Proof.* Let $C = \mathbf{x}_1, \mathbf{x}_2, \ldots, \mathbf{x}_n$ be a molecular conformation of the molecule $G$. For any two atoms $a_i$ and $a_j$ in $G$, their distance $d_{ij}$ is given by the Euclidean distance between their positions, $d_{ij} = |\mathbf{x}_i - \mathbf{x}_j|$. After applying the rigid transformation $T(\mathbf{R}, \mathbf{t})$, the new positions of the atoms are $\mathbf{x}'_i = \mathbf{R}\mathbf{x}_i + \mathbf{t}$ and $\mathbf{x}'_j = \mathbf{R}\mathbf{x}_j \mathbf{t}$. The distance between the transformed atom positions is $d'_{ij} = |\mathbf{x}'_i - \mathbf{x}'_j| = |\mathbf{R}\mathbf{x}_i + \mathbf{t} - (\mathbf{R}\mathbf{x}_j + \mathbf{t})| = |\mathbf{R}(\mathbf{x}_i - \mathbf{x}_j)|$. Since $\mathbf{R}$ is a rotation matrix from the group $SO(3)$, it preserves distances, and we have $|\mathbf{R}(\mathbf{x}_i - \mathbf{x}_j)| = |\mathbf{x}_i - \mathbf{x}_j|$. Thus, $d'_{ij} = d_{ij}$, and the relative arrangement of atoms in the molecule is preserved under the rigid transformation $T(\mathbf{R}, \mathbf{t})$. $\qquad\qquad\square$

One direct application of the link between stereochemistry and SE(3) geometry is the development of SE(3)-invariant molecular representations. These representations preserve the geometric information of the molecule while remaining invariant under SE(3) transformations, which can improve the performance of machine learning models for molecular property prediction.

### A.3    Implications of SE(3)-Invariance for Molecular Property Prediction

The interplay between stereochemistry and SE(3) geometry provides a theoretical foundation for the development of novel molecular representations and algorithms rooted in SE(3) geometry. This also provides a platform for designing innovative algorithms for molecular property prediction. Firstly, this connection suggests that embedding SE(3) geometric information into machine learning models can enhance their predictive capabilities regarding molecular properties. For instance, SE(3)-based graph neural networks can be employed to learn meaningful embeddings of molecules, taking into account their three-dimensional structure and chirality. Secondly, optimization algorithms that make use of SE(3) geometry can be engineered to explore for optimal molecular conformations or to align molecules in a way that acknowledges their stereochemical properties.

## B    Persistent Homology

In this section, we delve deeper into the concept of single parameter persistent homology. Broadly, the persistent homology (PH) procedure involves a three-step process.

The first phase, known as the *graph decomposition* stage, facilitates the incorporation of domain-specific information into the process. This procedure structures the data by creating a filtration sequence of simplicial complexes.

Subsequently, in the *persistence diagrams* stage, the mechanism records the emergence and disappearance (birth/death times) of topological features throughout the filtration sequence. This step meticulously documents the chronological evolution of these topological attributes.

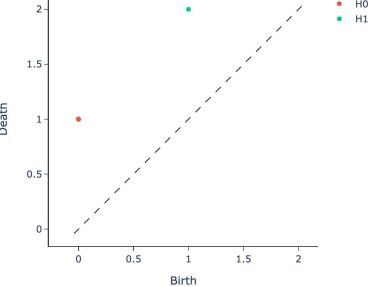

Figure 5: **Persistence Diagram** of Vietoris-Rips (VR) Filtration for Cytosine.

As an example in Figure 5, we capture the birth and death of topological features ($H_0$ and $H_1$) as the filtration progresses. Each dot represents a feature, with the x-coordinate denoting the 'birth' time (when the feature first appears), and the y-coordinate indicating the 'death' time (when the feature disappears). The $H_0$ points (connected components) typically appear along the diagonal,

reflecting that atoms (vertices) are born at the start of the filtration and die as they merge with other components. Conversely, $H_1$ points (1-dimensional holes) lie above the diagonal, arising when a cycle is formed and dying when it's filled in. The multiplicity of $H_0$ and $H_1$ points represents the number of connected components and holes, respectively, at different stages of the VR-filtration, providing insight into the molecule's topological complexity. In this example the multiplicity of $H_0$ is 12 and $H_1$ is 1.

The final stage involves *vectorization* or fingerprinting. In this step, the records generated in the persistence diagrams stage are transformed into a function or vector. These vectors are then ready for integration into appropriate machine learning models.

In essence, this three-step process transforms complex topological data into a simplified, machine-readable form, thereby enabling more effective analysis and predictions.

## B.1    Vietoris-Rips Filtration

Persistent Homology (PH) is a powerful tool for the chronological documentation of the topological transformations inherent in a sequence. The essence of this process lies in the assembly of the sequence, denoted as $\widehat{\mathcal{G}}_1 \subseteq \ldots \subseteq \widehat{\mathcal{G}}_N$. Herein, we leverage domain-specific data, such as atomic mass, partial charge, bond type and chirality, to enrich the PH sequence. The techniques employed in the creation of this PH sequence, also known as filtration methods, are manifold. However, we will primarily concentrate on two prevalent techniques: the *Sublevel/Superlevel filtration* and the *Vietoris-Rips (VR) filtration*.

Initiating with an unweighted graph, or compound, $\mathcal{G} = (\mathcal{V}, \mathcal{E})$ - where $\mathcal{V} = v_1, \ldots, v_m$ represents the nodes (atoms) and $\mathcal{E} = e_{rs}$ symbolizes the edges (bonds) - the frequently used approach deploys a filtration function $f : \mathcal{V} \to \mathbb{R}$. This function is coupled with a set of thresholds $\mathcal{I} = \alpha_i$ that follow the rule $\alpha_1 = \min_{v \in \mathcal{V}} f(v) < \alpha_2 < \ldots < \alpha_N = \max_{v \in \mathcal{V}} f(v)$.

For each $\alpha_i$ in $\mathcal{I}$, we identify a subset $\mathcal{V}_i$ that encompasses the nodes where the function value is less than or equal to the threshold, mathematically expressed as $\mathcal{V}_i = v_r \in \mathcal{V} \mid f(v_r) \leq \alpha_i$. The function $f$ may stand for a variety of domain-specific factors - from atomic mass and electron affinity to more abstract concepts like bond type and ionization energy. In addition, we could employ graph-based functions like node degree or betweenness centrality.

Following this, we conceive a series of nested subgraphs $\mathcal{G}_1 \subset \mathcal{G}_2 \subset \ldots \subset \mathcal{G}_N = \mathcal{G}$ that are induced by the subsets $\mathcal{V}_i$, such that $\mathcal{G}_i = (\mathcal{V}_i, \mathcal{E}_i)$ where $\mathcal{E}_i = e_{rs} \in \mathcal{E} \mid v_r, v_s \in \mathcal{V}_i$. To further the filtration process, each subgraph $\mathcal{G}_i$ is then affiliated with a simplicial complex $\widehat{\mathcal{G}}_i$.

One of the popular methods to do this is by assigning a $k$-simplex to every complete $(k+1)$-subgraph in $\mathcal{G}$. This method is known as the Sublevel filtration with clique complexes. Similarly, a Superlevel filtration can be obtained by changing the condition from $f(v_i) \leq \alpha_i$ to $f(v_i) \geq \alpha_i$.

For a graph equipped with weights (akin to bond strengths), the sublevel filtration on these weights would serve to encapsulate domain-specific information implicit in these weights. Although the Sublevel/Superlevel filtration with clique complexes is widely used due to its computational efficiency, our focus in this paper would be the more intricate Vietoris-Rips (VR) filtration, a distance-based method.

The VR filtration technique, albeit more computationally demanding, offers an in-depth understanding of the innate characteristics of a graph. Starting with a given graph $\mathcal{G} = (\mathcal{V}, \mathcal{E})$, we calculate the distance $d(v_r, v_s) = d_{rs}$ between each pair of nodes, defined as the minimum number of edges needed to travel from $v_r$ to $v_s$ in $\mathcal{G}$.

Subsequently, we construct a sequence of graphs, $\Gamma_n = (\mathcal{V}, \mathcal{E}_n)$, where $\mathcal{E}_n = e_{rs} \mid d_{rs} \leq n$, which signifies the addition of an edge for every pair of vertices with a distance less than or equal to $n$ in $\mathcal{G}$. Then, we build the simplicial complex $\Delta_n = \widehat{\Gamma}_n$ - the clique complex of $\Gamma_n$ - which results in a filtration $\Delta_0 \subset \Delta_1 \subset \cdots \subset \Delta_K$. Here, $K$ represents the maximum distance between any two nodes in the graph $\mathcal{G}$.

In essence, through these filtration techniques of PH, we can observe and comprehend the complex molecular structures in computational chemistry, thereby enhancing our ability to manipulate and exploit these structures for domain-specific applications.

## B.2  Persistence Diagrams

The next phase in the Persistent Homology (PH) process involves the construction of Persistence Diagrams (PDs) from the filtration sequence, denoted as $\Delta_0 \subset \Delta_1 \subset \cdots \subset \Delta_K$. As expounded earlier, PDs are an aggregation of 2-tuples, denoting the birth and death times of topological characteristics surfacing in the filtration, mathematically formulated as $\mathrm{PD}_k(\mathcal{G}) = (b_\sigma, d_\sigma) \mid \sigma \in H_k(\Delta_i)$ for $b_\sigma \leq i < d_\sigma$. This is a conventional step and numerous software libraries exist to simplify this task.

The subsequent phase involves the examination of an essential family of Stable Persistence (SP) vectorizations known as Persistence Curves. This term encompasses a variety of SP vectorizations like Betti Curves, Life Entropy, Landscapes, and others. Given that Persistence Curves generally yield a single-variable function, they can be expressed as 1D-vectors using an appropriate mesh size depending on the number of thresholds employed. The Multidimensional Persistence (MP) Fingerprint framework uses Betti Curves, one of the most commonly used Persistence Curves, to create multidimensional vectorizations. We will now elaborate on Betti Curves.

Betti curves offer a simplistic approach to SP vectorization as they quantify the presence of topological features at a given threshold interval. Specifically, $\beta_k(\Delta)$ signifies the total number of $k$-dimensional topological features in the simplicial complex $\Delta$, defined as $\beta_k(\Delta) = rank(H_k(\Delta))$. Subsequently, $\beta_k(\mathcal{G}) : [\epsilon_1, \epsilon_{q+1}] \to \mathbb{R}$ is a step function defined as $\beta_k(\mathcal{G})(t) = rank(H_k(\widehat{\mathcal{G}}_i))$ for $t \in [\epsilon_i, \epsilon_{i+1})$, where $\{\epsilon_i\}_1^q$ represents the thresholds for the filtration used.

Considering that this is a step function where the function remains constant for each interval $[\epsilon_i, \epsilon_{i+1})$, it can be accurately represented by a vector of size $1 \times q$ as $\vec{\beta}(\mathcal{G}) = [\beta(1)\ \beta(2)\ \beta(3)\ \dots\ \beta(q)]$.

With the threshold set $\{\beta_j\}_{j=1}^n$ for the second filtration function $g$, $\vec{\beta}_i = \vec{\beta}(PD(\mathcal{G}_i, g))$ will be a vector of size $1 \times n$. This means that for each $1 \leq i \leq m$, $\mathbf{M}_\beta^i = \vec{\beta}_i$ and the MP Betti Summary $\mathbf{M}_\beta(\mathcal{G})$ would be a 2D-vector (matrix) of size $m \times n$. Specifically, each entry $\mathbf{M}_\beta = [m_{ij}]$ is simply the Betti number of the corresponding clique complex in the bifiltration $\widehat{\mathcal{G}}_{ij}$, i.e., $m_{ij} = \beta(\widehat{\mathcal{G}}_{ij})$. This matrix $\mathbf{M}_\beta$, also referred to as the *bigraded Betti numbers* in the literature, offers computational advantages over other vectorizations, making it a preferred choice for many applications.

# C  Multiparameter Persistent Homology for Compound Fingerprinting

**Graph Decomposition:** Given an unweighted graph or compound $\mathcal{G} = (\mathcal{V}, \mathcal{E})$, where $\mathcal{V} = v_1, \dots, v_m$ denotes nodes (atoms) and $\mathcal{E} = e_{rs}$ denotes edges (bonds), we decompose $\mathcal{G}$ into subgraphs using a function $f : \mathcal{V} \to \mathbb{R}$ and threshold sets $\mathcal{I} = \{\alpha_i\}_{i=1}^m$. Here, $\alpha_1 = \min_{v \in \mathcal{V}} f(v) < \alpha_2 < \dots < \alpha_m = \max_{v \in \mathcal{V}} f(v)$. We then define $\mathcal{V}_i = v_r \in \mathcal{V} \mid f(v_r) \leq \alpha_i$, the sublevel sets for $f$, creating a hierarchy $\mathcal{V}_1 \subset \mathcal{V}_2 \subset \cdots \subset \mathcal{V}_m = \mathcal{V}$ among nodes relative to function $f$, resulting in a nested sequence of subgraphs (Figure 2). In molecular machine learning, $f$ can represent properties like atomic mass, partial charge, bond type, electron affinity, or ionization energy. Alternatively, $f$ can be graph-induced functions like node degree or betweenness.

**Constructing Vietoris-Rips (VR) Simplicial Complexes:** The first step involves calculating the shortest path distances (*geodesic distances*) between each node in the graph $\mathcal{G}$, denoted as $d(v_r, v_s) = d_{rs}$. With $K = \max d_{rs}$, we define a VR-filtration for each vertex set $\mathcal{V}_{i_0}$, yielding a sequence $\Delta_{i_0 0} \subseteq \Delta_{i_0 1} \subseteq \dots \subseteq \Delta_{i_0 K}$ (Figure 6). This results in $m \times (K+1)$ simplicial complexes $\Delta_{ij}$, forming the bipersistence module.

The bipersistence module can be viewed with $\mathcal{V}_i$ sequence increasing in the vertical direction, and induced VR-complexes $\Delta_{ij}$ in the horizontal direction. The slicing direction is fixed horizontally (VR-direction), from which persistence diagrams are derived.

A small graph $\mathcal{G}$ serves as a toy example in Figure 6. The sublevel filtration (vertical direction) uses the valency function, and each row develops a VR-filtration of the subgraph based on graph distances between nodes. With each subsequent column, edges are added based on nodes with an increasing geodesic distance, resulting in complete graphs in the final column.

Upon bifiltration completion, a single filtration $\mathcal{V}_{i_0} = \Delta_{i_0 0} \subseteq \Delta_{i_0 1} \subseteq \dots \subseteq \Delta_{i_0 K}$ is obtained for each $1 \leq i_0 \leq m$ in the horizontal direction. Each VR filtration threshold level provides a persistence

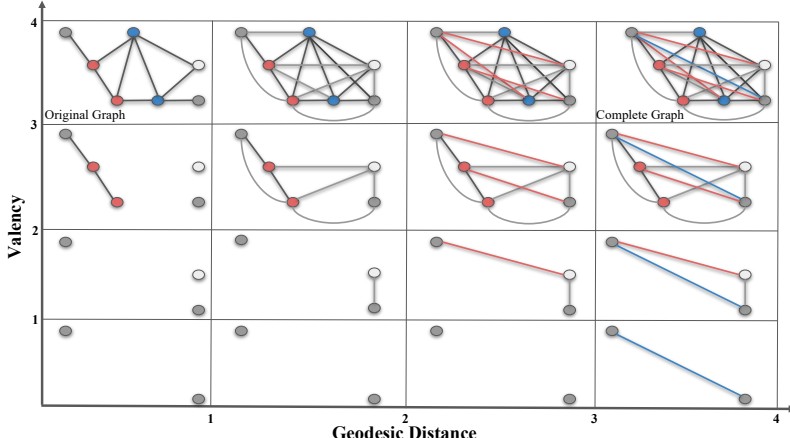

Figure 6: **Vietoris-Rips (VR) Simplicial Filtrations.** The illustration demonstrates a bifiltration of the graph $\mathcal{G}$ located in the top left corner, integrating a vertical sublevel filtration according to node degree (with valency thresholds of $1, 2, 3, 4$) and a horizontal Vietoris-Rips (VR) filtration based on geodesic length. The columns represent varying geodesic lengths: black edges for distance $\leq 1$, gray edges for distance $\leq 2$, red edges for distance $\leq 3$, and blue edges for distance $\leq 4$. The color of the nodes corresponds to their valency.

diagram $PD(\mathcal{V}_{i_0})$, yielding $m$ persistence diagrams $PD(\mathcal{V}_i)$. Applying a vectorization, $\varphi$, to each persistence diagram results in $m$ row vectors of fixed size $r$, i.e., $\vec{\varphi}i = \varphi(PD(\mathcal{V}_i))$. This generates a 2D-vector $\mathbf{M}\varphi$ of size $m \times (K+1)$.

**Persistence Diagrams:** Following VR filtration, we monitor the evolution of topological features in the simplicial complexes sequence $\{\widehat{G}_i\}_{i=1}^N$. These $k$-dimensional topological features could represent connected components (0-dimension), loops (1-dimension), or voids (2-dimension). Each $k$-dimensional topological feature $\sigma$ is associated with a unique pair $(b_\sigma, d_\sigma)$, where $1 \leq b_\sigma < d_\sigma \leq N$ represents the first appearance and disappearance of $\sigma$ in the filtration sequence, respectively. This pair is termed the birth time and death time of $\sigma$, with $d_\sigma - b_\sigma$ as its lifespan or persistence. Persistence diagrams capture these birth and death times of topological features. For any $0 \leq k \leq D$ (with $D$ as the highest dimension in the simplicial complex $\widehat{\mathcal{G}}_N$), the $k^{th}$ persistence diagram is defined as $\mathrm{PD}_k(\mathcal{G}) = (b_\sigma, d_\sigma) \mid \sigma \in H_k(\widehat{\mathcal{G}}_i)$ for $b_\sigma \leq i < d_\sigma$. Here, $H_k(\widehat{\mathcal{G}}_i)$ denotes the $k^{th}$ homology group of $\widehat{\mathcal{G}}_i$, which retains information about the $k$-holes in the simplicial complex $\widehat{\mathcal{G}}_i$. Our implementation utilizes 0 and 1 dimensional homology features, i.e., $PD_0(\mathcal{G})$ and $PD_1(\mathcal{G})$.

**Persistence Diagram Vectorizations (Fingerprinting):** While persistent homology reveals concealed patterns in data via persistence diagrams (sets of points in $\mathbb{R}^2$), these are not inherently suitable for statistical or machine learning applications. Persistence diagrams are commonly transformed into a format amenable to machine learning through kernels [37] or vectorizations [31]. This process essentially converts persistence diagrams into data fingerprints for machine learning usage. In our methodology, we adopt Betti curve vectorization [11] to translate the information encapsulated in persistent homology, represented as persistence diagrams, into a feature vector.

## D    Stability of Multiparameter Persistent Homology

A specific persistence diagram vectorization, denoted as $\varphi$, can be thought of as a mapping from the space of persistence diagrams to the space of functions. The concept of stability refers to the smoothness of this transformation. Essentially, it assesses whether a slight perturbation in the persistence diagram results in a significant change in the vectorization. To make this assessment meaningful, it is necessary to establish a metric in the space of persistence diagrams that defines what constitutes a "slight perturbation". The most commonly used metric for this purpose is the Wasserstein distance, also known as the matching distance, which is defined as follows.

Let $PD(\mathcal{X}^+)$ and $PD(\mathcal{X}^-)$ be persistence diagrams two datasets $\mathcal{X}^+$ and $\mathcal{X}^-$ (We omit the dimensions in PDs). Let $PD(\mathcal{X}^+) = \{q_j^+\} \cup \Delta^+$ and $PD(\mathcal{X}^-) = \{q_l^-\} \cup \Delta^-$ where $\Delta^\pm$ represents the diagonal (representing trivial cycles) with infinite multiplicity. Here, $q_j^+ = (b_j^+, d_j^+) \in PD(\mathcal{X}^+)$ represents the birth and death times of a topological feature $\sigma_j$ in $\mathcal{X}^+$. Let $\phi : PD(\mathcal{X}^+) \to PD(\mathcal{X}^-)$

represent a bijection (matching). With the existence of the diagonal $\Delta^\pm$ in both sides, we make sure the existence of these bijections even if the cardinalities $|\{q_j^+\}|$ and $|\{q_l^-\}|$ are different.

**Definition D.1.** *Let $PD(\mathcal{X}^\pm)$ be persistence diagrams of the datasets $\mathcal{X}^\pm$, and $\mathbf{M} = \{\phi\}$ represent the space of matchings as described above. Then, the $p^{th}$ Wasserstein distance $\mathcal{W}_p$ defined as*

$$\mathcal{W}_p(PD(\mathcal{X}^+), PD(\mathcal{X}^-)) = \min_{\phi \in \mathbf{M}} \left( \sum_j \|q_j^+ - \phi(q_j^+)\|_\infty^p \right)^{\frac{1}{p}}, \quad p \in \mathbb{Z}^+.$$

Now, let's define the stability of vectorizations. A vectorization can be viewed as a mapping from the space of persistence diagrams, $\mathbf{P}$, to the space of functions or vectors $\mathbf{Y}$, for example, $\Psi : \mathbf{P} \to \mathbf{Y}$. In particular, if $\Psi$ is the persistence landscape, then $\mathbf{Y} = \mathcal{C}([0, K], \mathbb{R})$ and if $\Psi$ is the Betti summary, then $\mathbf{Y} = \mathbb{R}^m$. The stability of the vectorization $\Psi$ refers to the continuity of $\Psi$ as a mapping. Let $\mathrm{d}(.,.)$ be a suitable metric on the space of vectorizations. The stability of $\Psi$ can then be defined as follows:

**Definition D.2.** *Let $\Psi : \mathbf{P} \to \mathbf{Y}$ be a vectorization for single persistence diagrams. Let $\mathcal{W}_p, \mathrm{d}$ be the metrics on $\mathbf{P}$ and $\mathbf{Y}$ respectively as described above. Let $\psi^\pm = \Psi(PD(\mathcal{X}^\pm)) \in \mathbf{Y}$. Then, $\Psi$ is called stable if*

$$\mathrm{d}(\psi^+, \psi^-) \leq C \cdot \mathcal{W}_{p_\Psi}(PD(\mathcal{X}^+), PD(\mathcal{X}^-))$$

In this context, the constant $C > 0$ is independent of $\mathcal{X}^\pm$. The stability inequality states that the changes in the vectorizations are limited by the changes in persistence diagrams. The proximity of two persistence diagrams is reflected in the proximity of their respective vectorizations. A vectorization $\varphi$ is referred to as *stable* if it satisfies this stability inequality for a given $\mathrm{d}$ and $\mathcal{W}_p$ [6].

Now, we are ready to show the stability of MPPH Fingerprints. Consider two graphs, $\mathcal{G}^+ = (\mathcal{V}^+, \mathcal{E}^+)$ and $\mathcal{G}^- = (\mathcal{V}^-, \mathcal{E}^-)$. A stable single parameter persistence vectorization is represented by $\varphi$, and it satisfies the stability equation,

$$\mathrm{d}(\varphi(PD(\mathcal{G}^+)), \varphi(PD(\mathcal{G}^-))) \leq C_\varphi \cdot \mathcal{W}_{p_\varphi}(PD(\mathcal{G}^+), PD(\mathcal{G}^-)) \tag{2}$$

for some $1 \leq p_\varphi \leq \infty$. Here, $\varphi(\mathcal{G}^\pm)$ represent the corresponding vectorizations for $PD(\mathcal{G}^\pm)$ and $\mathcal{W}_p$ represents Wasserstein-$p$ distance as defined in Definition 4.1.

Now, let $f : \mathcal{V}^\pm \to \mathbb{R}$ be a filtration function with threshold set $\{\alpha_i\}_{i=1}^m$. Then, define the sublevel vertex sets $\mathcal{V}_i^\pm = \{v_r \in \mathcal{V}^\pm \mid f(v_r) \leq \alpha_i\}$. For each $\mathcal{V}_i^\pm$, construct the induced VR-filtration $\Delta_{i0}^\pm \subset \Delta_{i1}^\pm \subset \cdots \subset \Delta_{iK}^\pm$ as before. For each $1 \leq i_0 \leq m$, we will have persistence diagram $PD(\mathcal{V}_{i_0}^\pm)$ of the filtration $\{\Delta_{i_0 k}^\pm\}$.

The induced matching distance between multiple persistence diagrams is defined as follows,

$$\mathbf{D}_{p,f}(\mathcal{G}^+, \mathcal{G}^-) = \sum_{i=1}^m \mathcal{W}_p(PD(\mathcal{V}_i^+), PD(\mathcal{V}_i^-)). \tag{3}$$

Now, we define the distance between induced MPPH Fingerprints as,

$$\mathfrak{D}_f(\mathbf{M}_\varphi(\mathcal{G}^+), \mathbf{M}_\varphi(\mathcal{G}^-)) = \sum_{i=1}^m \mathrm{d}(\varphi(PD(\mathcal{V}_i^+)), \varphi(PD(\mathcal{V}_i^-))) \tag{4}$$

**Theorem D.1.** *Let $\varphi$ be a stable single parameter persistence vectorization. Then, the induced MPPH Fingerprint $\mathbf{M}_\varphi$ is also stable, i.e., with the notation above, there exists $\widehat{C}_\varphi > 0$ such that for any pair of graphs $\mathcal{G}^+$ and $\mathcal{G}^-$, we have the following inequality.*

$$\mathfrak{D}(\mathbf{M}_\varphi(\mathcal{G}^+), \mathbf{M}_\varphi(\mathcal{G}^-)) \leq \widehat{C}_\varphi \cdot \mathbf{D}_{p_\varphi}(\{PD(\mathcal{G}^+)\}, \{PD(\mathcal{G}^-)\})$$

*Proof:* Given $\varphi$'s stability by Equation 2, for every $1 \leq i \leq m$, a certain constant $C_\varphi > 0$ makes the following true: $\mathrm{d}(\varphi(PD(\mathcal{V}_i^+)), \varphi(PD(\mathcal{V}_i^+))) \leq C_\varphi \cdot \mathcal{W}_{p_\varphi}(PD(\mathcal{V}_i^+), PD(\mathcal{V}_i^-))$, where $\mathcal{W}_{p_\varphi}$ denotes the Wasserstein-$p$ distance. With this, we can simplify the distance between MPPH

Fingerprints of two graphs as:

$$
\begin{aligned}
\mathfrak{D}(\mathbf{M}_\varphi(\mathcal{G}^+), \mathbf{M}_\varphi(\mathcal{G}^-)) \quad &= \quad \sum_{i=1}^{m} \mathrm{d}(\varphi(PD(\mathcal{V}_i^+)), \varphi(PD(\mathcal{V}_i^-))) \\
&\leq \quad \sum_{i=1}^{m} C_\varphi \cdot \mathcal{W}_{p_\varphi}(PD(\mathcal{V}_i^+), PD(\mathcal{V}_i^-)) \\
&= \quad C_\varphi \sum_{i=1}^{m} \mathcal{W}_{p_\varphi}(PD(\mathcal{V}_i^+), PD(\mathcal{V}_i^-)) \\
&= \quad C_\varphi \cdot \mathbf{D}_{p_\varphi}(\mathcal{G}^+, \mathcal{G}^-)
\end{aligned}
$$

where the first and last equalities are due to Equation 3 and Equation 4 respectively, while the inequality follows from Equation 2, which is valid for any $i$.

## E  Computational Complexity of MPPH Fingerprints

The computational complexity (CC) of producing the MPPH Fingerprint, denoted as $\mathbf{M}_\psi^d$, hinges on two factors: the vectorization method, symbolized by $\psi$, and the number of filtration functions, represented by $d$. For single parameter persistence, the worst case CC is $\mathcal{O}(\mathcal{N}^3)$, where $\mathcal{N}$ is the count of $k$-simplices [62]. This complexity arises from the cubic time complexity of Gaussian elimination, employed to simplify the boundary matrix that encodes the simplicial complex [17]. However, multiparameter persistence is inherently more complex. The CC for the MPPH Fingerprint is $\mathcal{O}(d \cdot r \cdot \mathcal{N}^3 \cdot C_\psi(m))$. Here, $r$ stands for the resolution of the multipersistence grid, and $C_\psi(m)$ indicates the CC of the chosen vectorization technique, with $m$ being the number of barcodes in our $k$-dimensional persistence diagram. For instance, if we use the persistence landscape as $\psi$, $C_\psi(m)$ equals $m^2$. Hence, for a multiparameter persistence landscape with four filtration functions, the CC would be $\mathcal{O}(4 \cdot r \cdot \mathcal{N}^3 \cdot m^2)$. However, Betti curve vectorization simplifies the computation. It bypasses the need for persistence diagrams and concentrates on calculating the rank of homology groups in the multiparameter persistence module. Utilizing minimal representations significantly reduces the CC for Betti vectorization. To further optimize the process, we employ parallel processing across an Intel Core i7 CPU's eight cores, supported by 100 GB of RAM. More specifics regarding the computational time for MPPH Fingerprint generation from different datasets are presented in G.2. Compared to graph-based models like those identifying common molecular fragments or motifs, MPPH requires fewer computational resources during the training phase.

## F  Uncertainty Quantification

In drug discovery and materials science, accurate prediction of molecular properties is crucial for decision-making processes, and gauging the level of uncertainty provides valuable information about the reliability and confidence of the predictions. This enables researchers to make well-informed decisions, assess risks, efficiently allocate resources, and improve model performance. For instance, in the context of drug discovery, if a candidate molecule exhibits high uncertainty in its predicted bioactivity, it may warrant additional experimental validation before proceeding to expensive and time-intensive developmental stages. The benefits of incorporating uncertainty estimation in molecular property prediction include:

1. Model evaluation and selection: Uncertainty estimation can help assess the performance of different models/algorithms by comparing their uncertainties. A model with lower uncertainty in its predictions is generally preferred, as it indicates higher confidence in the predicted properties.

2. Active learning and data acquisition: Uncertainty estimation can guide active learning strategies by identifying the most informative data points to acquire next. By prioritizing data points with high uncertainty, researchers can efficiently allocate resources to experiments that are expected to provide the most significant improvements in model performance.

3. Interpretability and understanding of model limitations: Uncertainty estimation can help researchers understand the limitations of their models, such as areas where the model might

not generalize well due to inherent class overlap or measurement noise (*aleatoric uncertainty*) or where it lacks sufficient training data (*episdemic uncertainty*). This understanding can lead to the development of better models or the identification of areas that require more focused experimental data collection.

### F.1 Uncertainty Quantification via SGLB Ensembles

The gradient boosting algorithm [21] iteratively builds a model $F$ to minimize the empirical risk $L(F|D)$, where $D$ represents the dataset and $L$ is a loss function. This model is updated at each step $t$, given by:

$$F^{(t)}(x) = F^{(t-1)}(x) + \eta h^{(t)}(x) \tag{5}$$

In this equation, $F^{(t-1)}$ denotes the model from the last step, $h^{(t)}(x)$ is a weak learner selected from a function family $\mathcal{H}$, and $\eta$ symbolizes the learning rate. The weak learner $h^{(t)}$ is typically chosen to approximate the negative gradient $-g^{(t)}(x, y)$, as per:

$$h^{(t)} = \arg\min_{h \in \mathcal{H}} \mathbb{E}_D \left[ -g^{(t)}(x, y) - h(x) \right]^2 \tag{6}$$

When considering an ensemble of probabilistic models $P(y|x; \theta^{(m)})_{m=1}^{M}$ drawn from the posterior $p(\theta|D)$, each model $P(y|x, \theta^{(m)})$ offers a unique estimate of data uncertainty, signified by the entropy of its predictive distribution. Knowledge uncertainty is expressed as the level of variance, or "disagreement", among models in the ensemble.

Exact Bayesian inference is often unfeasible, leading to the use of an explicit or implicit approximation $q(\theta)$ to the true posterior $p(\theta|D)$. While various approximations have been investigated for neural network models, the exploration of Bayesian inference for gradient-boosted trees remains less explored.

Considering $p(\theta|D)$, the predictive posterior of the ensemble is calculated by taking the expectation concerning the models in the ensemble:

$$P(y|x, D) = \mathbb{E}_{p(\theta|D)} P(y|x; \theta) \approx \frac{1}{M} \sum_{m=1}^{M} P(y|x; \theta^{(m)}), \quad \theta^{(m)} \sim p(\theta|D) \tag{7}$$

The entropy of the predictive posterior estimates total uncertainty:

$$H[P(y|x, D)] = \mathbb{E}_{P(y|x,D)} - \ln P(y|x, D) \tag{8}$$

Total uncertainty emerges from both data uncertainty and knowledge uncertainty. In applications like active learning and out-of-domain detection, it is beneficial to estimate knowledge uncertainty separately. The sources of uncertainty can be decomposed by considering the mutual information between the parameters $\theta$ and the prediction $y$:

$$I[y, \theta|x, D] = H[P(y|x, D)] - \mathbb{E}_{p(\theta|D)} H[P(y|x; \theta)] \tag{9}$$

where $I[y, \theta|x, D]$ represents knowledge uncertainty, $H[P(y|x, D)]$ denotes total uncertainty, and $\mathbb{E}_{p(\theta|D)} H[P(y|x; \theta)]$ signifies expected data uncertainty.

This can be approximated by:

$$\approx H \left[ \frac{1}{M} \sum_{m=1}^{M} P(y|x; \theta^{(m)}) \right] - \frac{1}{M} \sum_{m=1}^{M} H[P(y|x; \theta^{(m)})] \tag{10}$$

In the case of ensembles of probabilistic regression models $p(y|x; \theta^{(m)})_{m=1}^{M}$ over a continuous-valued target $y \in \mathbb{R}$, tractable estimates of the entropy of the predictive posterior, and consequently, mutual

information, are not achievable. Here, uncertainty estimates can be alternatively derived using the law of total variance:

$$V_{p(y|x,D)}[y] = V_{p(\theta|D)}\mathbb{E}_{p(y|x,\theta)}[y] + \mathbb{E}_{p(\theta|D)}V_{p(y|x,\theta)}[y] \tag{11}$$

where $V_{p(y|x,D)}[y]$ signifies total uncertainty, $V_{p(\theta|D)}\mathbb{E}p(y|x,\theta)[y]$ denotes knowledge uncertainty, and $\mathbb{E}p(\theta|D)Vp(y|x,\theta)[y]$ symbolizes expected data uncertainty.

Knowledge uncertainty can be estimated by evaluating an ensemble of models $p(y|x;\theta^{(m)})_{m=1}^{M}$ drawn from the posterior $p(\theta|D)$. The degree of variation or "disagreement" among the models serves as an estimate of knowledge uncertainty. One method to create an ensemble is to consider multiple independent models produced via Stochastic Gradient Langevin Boosting (SGLB). SGLB merges gradient boosting with stochastic gradient Langevin dynamics to achieve convergence to the global optimum even for non-convex loss functions. Initially, Gaussian noise is directly injected into the gradients, replacing the earlier weak learner equation with:

$$h^{(t)} = \arg\min_{h \in \mathcal{H}} \mathbb{E}_D \left[ -g^{(t)}(x,y) - h(x,\phi) + \nu \right]^2, \quad \nu \sim \mathcal{N}\left( 0, \frac{2}{\beta\eta}\mathbb{I}_{|\mathcal{D}|} \right) \tag{12}$$

Here, $\beta$ is the inverse diffusion temperature and $\mathbb{I}_{|\mathcal{D}|}$ is an identity matrix. This random noise $\nu$ aids in the exploration of the solution space to find the global optimum, and the diffusion temperature manages the level of exploration. $\phi$ denotes a set of parameters in the model. Specifically, it's used in the weak learner $h^{(t)}(x,\phi^{(t)})$ to denote the parameters at iteration $t$. This indicates that the weak learner at each iteration can be parameterized differently, enabling the model to learn and adapt over time.

Secondly, the previous update equation is modified as:

$$F^{(t)}(x) = (1\gamma\eta)F^{(t-1)}(x) + \eta h^{(t)}(x,\phi^{(t)}) \tag{13}$$

In this equation, $\gamma$ represents a regularization parameter. This alteration introduces a regularization term that moderates the influence of the preceding model $F^{(t-1)}(x)$ on the present model $F^{(t)}(x)$, and the weak learner $h^{(t)}(x,\phi^{(t)})$ is now contingent on the parameters $\phi^{(t)}$. These modifications in SGLB offer a more resilient and globally optimal solution compared to the conventional Gradient Boosting.

### F.2 Macro Setup

We employed the CatBoost gradient boosting library to construct distinct classification and regression models. The classification model was configured with 1,000 iterations, a maximum tree depth of 6, a learning rate of 0.05, and used the Logloss loss function for binary classification. For reproducibility, we set the random seed to 0. The regression model was similarly configured, with the RMSEWithUncertainty as the loss function. Both models' parameter settings were meticulously optimized using the Optuna hyperparameter optimization framework [3]. This facilitated a thorough exploration of the hyperparameter space, enabling us to find the optimal combination for our models.

## G  Further Experimental Details

### G.1  Dataset Statistics

**Lipophilicity**[1] is a crucial characteristic of drug molecules that impacts both permeability through membranes and solubility. The dataset, sourced from the ChEMBL database, contains experimental results for the octanol/water distribution coefficient (logD at pH 7.4) of 4200 compounds.

**FreeSolv**[2] is a compilation of both calculated and experimentally determined hydration free energies for 642 small molecules in water.

---

[1]https://deepchemdata.s3-us-west-1.amazonaws.com/datasets/Lipophilicity.csv:
[2]https://deepchemdata.s3.us-west-1.amazonaws.com/datasets/freesolv.csv.gz:

**ESOL**[3] is a collection of 1128 chemical compounds and their corresponding water solubility values.

**BACE**[4] offers quantitative IC50 values and qualitative binding results for 1513 human $\beta$-secretase 1 (BACE-1) inhibitors. Among these inhibitors, 691 are classified as active, while 822 are inactive.

**BBBP**[5] contains binary labels for 2050 compounds, aims to model and predict blood-brain barrier (BBB) permeability, a crucial factor in developing drugs targeting the central nervous system. Among these compounds, 1567 are capable of penetrating the blood-brain barrier, while 483 are not.

**ClinTox**[6] contains 1479 drugs, each annotated with two binary labels indicating clinical trial toxicity prediction and FDA approval status. Among these drugs, 1367 have displayed no evidence of toxicity in clinical trials, while 112 have been identified as toxic.

### G.2 Computation Time

The clock time performance for extracting Vietoris-Rips persistent homology features can be optimized by distributing the process across multiple CPU cores. We utilize parallelization techniques and allocate computational resources to 8 cores of a single Intel Core i7 CPU to improve extraction speed. Efficient algorithms, optimized data structures, and specialized libraries can further enhance the performance of extracting Vietoris-Rips persistent homology features.

Table 3: The clock time performance of extracting multiparameter Vietoris-Rips persistent homology features.

| Dataset | Atomic Mass | Partial Charge | Bond Type | Chirality | All Parameters |
|---|---|---|---|---|---|
| Lipophilicity | 55 sec | 42 sec | 17 sec | 13 sec | 127 sec |
| FreeSolv | 7 sec | 7 sec | 2 sec | 2 sec | 18 sec |
| ESOL | 11 sec | 11 sec | 4 sec | 3 sec | 29 sec |
| BACE | 20 sec | 15 sec | 6 sec | 5 sec | 46 sec |
| BBBP | 27 sec | 21 sec | 8 sec | 6 sec | 62 sec |
| ClinTox | 19 sec | 15 sec | 6 sec | 5 sec | 45 sec |

### G.3 Further Evaluations on Imbalanced Datasets

PRC-AUC is the area under the curve of Precision($q$) vs Recall($q$) for $q \in [0, 1]$, where Precision($q$) and Recall($q$) are defined as follows:

$$\text{Precision}(q) = \frac{\text{TP}(q)}{\text{TP}(q) + \text{FP}(q)} \quad \text{and} \quad \text{Recall}(q) = \frac{\text{TP}(q)}{\text{TP}(q) + \text{FN}(q)}$$

where TP($q$), FP($q$), FN($q$) are the weights of the true positive, false positive, and false negative samples, respectively.

Let's denote $p_i$ as the predicted probability that the $i^{th}$ instance belongs to the positive class. The actual class of this instance is represented by $c_i$. The weight of each instance, potentially used for differential weighting during PRC-AUC computation, is represented by $w_i$. In the simplest scenario we employ, all weights are set equal to 1, which signifies that each instance contributes equally to the computation. For a binary classification model, the PRC-AUC calculation can be expressed as $TP(q) = \sum_i w_i [p_i > q] c_i$. For a multi-classification model, we define positive samples as those belonging to class 0, with all others being negative. The true positive rate in this scenario is expressed as $TP(q) = \sum_i w_i [p_{i0} > q][c_i = 0]$.

The F1 score is the harmonic mean of precision and recall,

$$F1 = 2 \times \frac{\text{Precision} \times \text{Recall}}{\text{Precision} + \text{Recall}}$$

The F1 score balances the trade-off between precision and recall, especially in cases where the data suffer from class imbalance. This is highly relevant in our datasets: BACE, BBBP, and ClinTox.

---

[3]`https://deepchemdata.s3-us-west-1.amazonaws.com/datasets/delaney-processed.csv`

[4]`https://deepchemdata.s3-us-west-1.amazonaws.com/datasets/bace.csv`

[5]`https://deepchemdata.s3-us-west-1.amazonaws.com/datasets/BBBP.csv`

[6]`https://deepchemdata.s3-us-west-1.amazonaws.com/datasets/clintox.csv.gz`

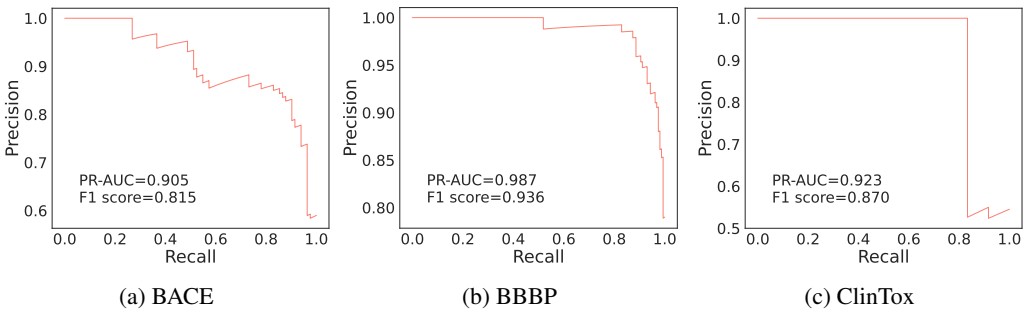

(a) BACE           (b) BBBP           (c) ClinTox

Figure 7: **Classification Performance on Imbalanced Datasets.**

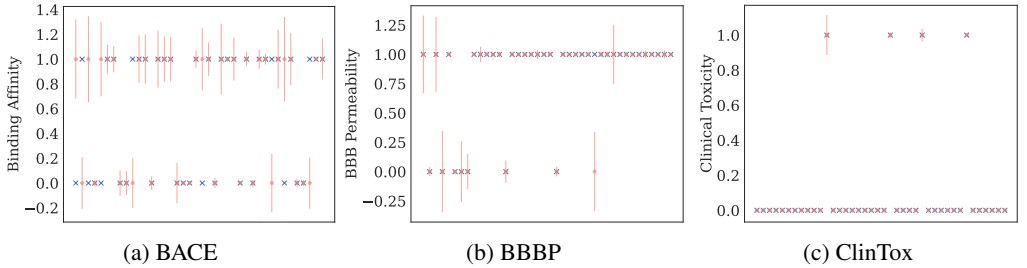

(a) BACE           (b) BBBP           (c) ClinTox

Figure 8: **Model Predictions and Uncertainty Visualization.** This figure displays the model's prediction results (represented by red circle markers), total uncertainty (encompassing both epistemic and aleatoric uncertainty, illustrated by vertical bars), and the actual labels (denoted by blue 'X' marks).

Figure 7 presents the PR curves, PRC-AUC scores, and F1 scores for three imbalanced datasets: BACE, BBBP, and ClinTox. The PR curves provide a visual representation of the trade-off between precision and recall for each model at various threshold settings. The PRC-AUC scores offer a single-value summary of the models' performances across all thresholds, with a higher score indicating a better model. The F1 scores are the harmonic mean of precision and recall, providing a balanced measure of model performance, especially critical for these imbalanced datasets. The results illustrate our models' ability to handle imbalance effectively, thus facilitating reliable predictions in real-world scenarios where data imbalance is prevalent.

### G.4 Further Evaluations on Uncertainty Quantification

Figure 8 illustrates the uncertainty ranges associated with misclassified samples and their patterns within highly imbalanced datasets, specifically BBBP and ClinTox. In the case of misclassified samples, the uncertainty bars are consistently broad, highlighting the model's awareness of its incorrect predictions. Furthermore, a pronounced disparity is observed in the extent of uncertainty between the samples of the minority and majority classes. In particular, samples from the minority class exhibit substantially larger uncertainty compared to those from the majority class. This pattern is predominantly driven by epistemic (data) uncertainty, a consequence of data sparsity inherent in minority class samples. The dominance of epistemic uncertainty in these scenarios is a clear testament to the efficacy of the SGLB model in providing reliable uncertainty quantification, especially in the challenging setting of imbalanced datasets.

## H Societal Impact and Limitations

### H.1 Social Impact

The ability to predict molecular properties through computational methods has vast societal implications in drug discovery. Accurate molecular property prediction can expedite the process of finding effective and safe drugs, consequently saving lives and reducing healthcare costs. It enables researchers to anticipate how a potential drug will interact with its target, forecast its possible side effects, and predict its pharmacokinetic (often summarized by the ADME properties of a drug: Ab-

sorption, Distribution, Metabolism, and Excretion) and pharmacodynamic (the relationship between drug concentrations/dose and pharmacologic or toxicologic responses) properties. In this context, the relevance of molecular property prediction for in silico experiments and high throughput screening is significant. In silico experiments allow researchers to perform virtual simulations to predict how a drug might perform in a biological system, substantially reducing the time and cost associated with physical experiments. They also facilitate high throughput screening, where thousands of potential drugs can be tested simultaneously for a specific property, substantially increasing the efficiency of drug discovery. Moreover, uncertainty quantification in these predictions is crucial. It provides a measure of confidence in the predictions, informing researchers about the reliability of their models and assisting in risk assessment. This level of transparency is vital in making informed decisions about further investigations and resources allocation.

## H.2 Limitations

Our model's computational characteristics, detailed in Section E, showcase its adaptability and potential scalability for large libraries. However, a primary limitation arises with the computation time of multiparameter persistent homology fingerprints, particularly when tackling ultra-large scale compound libraries (comprising millions to billions of compounds). This issue emphasizes the necessity of optimizing the allocation of computational resources for such expansive datasets.

To mitigate this limitation, additional CPU cores on a High-Performance Computing (HPC) platform can be used to parallelize the most computationally intensive operations, such as VR-filtration. Moreover, optimization of array operations (e.g., numpy) is achievable via the joblib library, further enhancing the model's computational efficiency.

When contrasted with alternatives, our model's performance is especially commendable for smaller scale compound libraries. It significantly outperforms variants of MPNNs in terms of accuracy as demonstrated in Table 1. Additionally, multiparameter persistent homology generally requires fewer computational resources during training than current graph-based models, which encode a compound by mining common molecular fragments or motifs, as discussed in [33]. For instance, training a motif-based Graph Neural Network (GNN) on the GuacaMol dataset, containing roughly 1.5M drug-like molecules, necessitates about 130 GPU hours [44]. We show the execution time of our computation pipeline in Table 3, when feature extraction tasks are distributed across 8 cores of a single Intel Core i7 CPU.

In conclusion, despite computational challenges when dealing with larger scale libraries, our model provides robust, efficient solutions, particularly for smaller scale compound libraries, reinforcing its value in computational chemistry and drug discovery.

