# OpenReview forum: "SE(3)-Invariant Multiparameter Persistent Homology for Chiral-Sensitive Molecular Property Prediction"
_NeurIPS.cc/2023/Workshop/AI4Science — NeurIPS2023-AI4Science Poster_

### Official Review · Reviewer_DoFi · 2023-10-09
**A careful study demonstrating the importance of domain specificity and robust confidence measures**

**Rating:** 7
**Confidence:** 3

**Review:**

1. **Summary**: The authors propose a method to predict the properties of molecules with a chirality-sensitive (i.e., SE(3)-invariant) persistent homology algorithm. The results of the authors' experiments seem to suggest that:
    * (I.) SE(3) invariance is theoretically and empirically important to model for accurate molecular property predictions; and that
    * (II.) Gradient boosting methods outperform state-of-the-art graph neural networks (GNNs) for molecular property prediction (regarding the MoleculeNet benchmarks).

   Overall, these results are interesting findings for the field and should enable future works to build upon these findings in the context of molecular property prediction and beyond (e.g., for building new chirality-sensitive GNN architectures).
2. **Strengths and Weaknesses**:
   * Points of strength:
     - The authors' experiments are well-executed and carefully designed.
     - The authors highlight the importance of achieving chirality sensitivity for molecular property predictions through not only motivating real-world examples (e.g., in Figure 1) but also through their empirical results in the main text and in the appendix (e.g., Table 3, chirality ablations).
     - The authors' proposed gradient boosting method clearly outperforms all baseline comparison methods, including state-of-the-art GNNs for molecular tasks.
     - The authors include a plethora of additional experiments and discussions that nicely complement the main text and its presented results.
   * Points for improvement:
     - My main concern is that the main text reads much like an appendix would for SE(3) equivariance proofs in a standard 3D graph neural network paper. I can appreciate the authors' detailed derivations here (especially for a full conference submission), but for a workshop submission, this may serve as a negative aspect rather than a positive aspect regarding its presentation. I might recommend simplifying the presentation of the main text (in the context of this workshop submission) so that workshop readers can more clearly and directly grasp the main messages of this work, which (in my view) are (1) the importance of SE(3) invariance/chirality sensitivity and (2) the benefits of gradient boosting methods over GNNs when combining them with domain-specific regression methods for molecular property prediction.
     - Additionally, I would recommend the authors add a brief discussion on the presence of 2D or 3D graph neural networks that achieve sensitivity to chirality through either filtration-like functions or 3D geometric techniques (e.g., ClofNet, GCPNet, and LEFTNet).
3. **Recommendation**: Given the authors' interesting and promising efforts in achieving highly effective and chirality-sensitive molecular property predictions, I am inclined to **accept** this work.
4. **Rationale behind Recommendation**: Given the novelty of the authors' proposed method (i.e., achieving invariance to the SE(3) group with multiparameter persistent homology), I think a score of 7 for this work is fair.
5. **Questions**:
   (1) Have the authors considered evaluating how well their proposed chirality-sensitive filtration function ($f$) may enhance existing 2D or 3D graph neural networks for molecular property prediction? In other words, does such a function also improve the performance of other methods? If so, I think that would be an interesting finding on its own.
6. **Feedback**: I personally appreciate the authors' efforts in carefully explaining the importance of recognizing chirality in one's molecular prediction methods, as such work is relatively scarce in the literature to date.
7. **Submission Type**: The authors' manuscript successfully complies with the four to eight-page requirement for the workshop's submissions. Great work!

---

### Official Review · Reviewer_7Wuk · 2023-10-13
**SE(3)-Invariant Fingerprint for Molecular Property Prediction**

**Rating:** 7
**Confidence:** 3

**Review:**

The paper proposes an SE(3) invariant molecular fingerprint generated using Vietoris-Rips persistent homology. The proposed fingerprint captures chirality and is stable. The authors also quantify uncertainties using a gradient boosting algorithm, and use active learning to reduce training cost. On MoleculeNet benchmark datasets, the proposed method achieves higher prediction accuracy than baselines over a range of molecular properties.

The paper is overall clear, and empirical results are strong. However, the main text does not contain enough theoretical background for a general audience, and several results in the experiment section can be explained more clearly.

### Pros
-	Fingerprints that distinguish chirality are well motivated. The theoretical guarantee of SE(3) invariance appears sound. The proposed method also satisfies a stability property, which is a desirable property for fingerprints.
-	The experiments are comprehensive, covering state-of-the-art GNN baselines and a variety of molecular property prediction tasks. The improvement over baselines are significant.


### Cons
-	The theorems in the main text contain a number of undefined notations and missing definitions, e.g. $PH$, $VR$, and a definition for Vietoris-Rips filtration. Consequently, the main text is not self-contained and is less accessible to a machine learning audience. Perhaps some proofs can be deferred to the appendix to make space for the basic definitions.
-	The importance of theorem 4.3 could be elaborated further. Why is stability important? Do existing methods have similar stability guarantees? It may also help to explain how the MPPH fingerprints are obtained somewhere in the main text.
-	The Bayesian ensemble seems to used only with MPPH. Can incorporating SGLB help improve the baselines too?
-	In line 259-363, the authors state that MPPH can incorporating bond polarity to augment the performance of lipophilicity prediction, yet MPPH’s MSE error of lipophilicity is not strong.
-	In addition to MSE, metrics that compare distributions may help demonstrate the advantage of uncertainty estimation beyond its usage in active learning.
-	The wording on line 125 appears to reveal the identity of the authors.

---

### Meta-Review · Area_Chair_vuHt · 2023-10-27

**Recommendation:** Accept (Oral)
**Confidence:** 3

**Metareview:**

The paper presents a method for predicting molecular properties using an SE(3)-invariant molecular fingerprint generated via Vietoris-Rips persistent homology. The approach incorporates chirality and quantifies uncertainties using gradient boosting, with active learning to reduce training cost. The empirical results show improved prediction accuracy over baselines on MoleculeNet benchmark datasets. Reviewers appreciate the theoretical foundation but suggests clarifying notations and definitions, explaining the importance of stability, and providing more details on how MPPH fingerprints are obtained. Moreover, reviewers highlight the significance of SE(3) invariance and the superiority of gradient boosting, recommending simplification of the main text and discussion of related work.
The experiments are well-executed and comprehensive. While there are suggestions for improving clarity and presentation, the contributions are significant and suitable for the workshop.